# Alterations in Circulating Monocytes Predict COVID-19 Severity and Include Chromatin Modifications Still Detectable Six Months after Recovery

**DOI:** 10.3390/biomedicines9091253

**Published:** 2021-09-17

**Authors:** Alberto Utrero-Rico, Cecilia González-Cuadrado, Marta Chivite-Lacaba, Oscar Cabrera-Marante, Rocío Laguna-Goya, Patricia Almendro-Vazquez, Carmen Díaz-Pedroche, María Ruiz-Ruigómez, Antonio Lalueza, María Dolores Folgueira, Enrique Vázquez, Ana Quintas, Marcos J. Berges-Buxeda, Moisés Martín-Rodriguez, Ana Dopazo, Antonio Serrano-Hernández, José María Aguado, Estela Paz-Artal

**Affiliations:** 1Instituto de Investigación Sanitaria 12 de Octubre (imas12), 28041 Madrid, Spain; ceciliagcuadrado@gmail.com (C.G.-C.); marta.chivite@gmail.com (M.C.-L.); oscar.cabrera@salud.madrid.org (O.C.-M.); rociolagunagoya@gmail.com (R.L.-G.); patricia.almendro.vazquez@gmail.com (P.A.-V.); cdiazp@salud.madrid.org (C.D.-P.); mrruigomez@salud.madrid.org (M.R.-R.); lalueza@hotmail.com (A.L.); mfolgueira@salud.madrid.org (M.D.F.); marcosjb@ucm.es (M.J.B.-B.); moises.martin.rodriguez@alumnos.upm.es (M.M.-R.); aserranoh@gmail.com (A.S.-H.); jaguadog1@gmail.com (J.M.A.); estela.paz@salud.madrid.org (E.P.-A.); 2Department of Immunology, Hospital Universitario 12 de Octubre, 28041 Madrid, Spain; 3Department of Internal Medicine, Hospital Universitario 12 de Octubre, 28041 Madrid, Spain; 4Department of Microbiology, Hospital Universitario 12 de Octubre, 28041 Madrid, Spain; 5Genomics Unit, Centro Nacional de Investigaciones Cardiovasculares (CNIC), 28029 Madrid, Spain; enrique.vazquez@cnic.es (E.V.); ana.quintas@cnic.es (A.Q.); adopazo@cnic.es (A.D.); 6Unit of Infectious Diseases, Hospital Universitario 12 de Octubre, 28041 Madrid, Spain

**Keywords:** circulating monocytes, COVID-19, HLA-DR, transcriptome, chromatin accessibility

## Abstract

An early analysis of circulating monocytes may be critical for predicting COVID-19 course and its sequelae. In 131 untreated, acute COVID-19 patients at emergency room arrival, monocytes showed decreased surface molecule expression, including low HLA-DR, in association with an inflammatory cytokine status and limited anti-SARS-CoV-2-specific T cell response. Most of these alterations had normalized in post-COVID-19 patients 6 months after discharge. Acute COVID-19 monocytes transcriptome showed upregulation of anti-inflammatory tissue repair genes such as *BCL6*, *AREG* and *IL-10* and increased accessibility of chromatin. Some of these transcriptomic and epigenetic features still remained in post-COVID-19 monocytes. Importantly, a poorer expression of surface molecules and low *IRF1* gene transcription in circulating monocytes at admission defined a COVID-19 patient group with impaired SARS-CoV-2-specific T cell response and increased risk of requiring intensive care or dying. An early analysis of monocytes may be useful for COVID-19 patient stratification and for designing innate immunity-focused therapies.

## 1. Introduction

More than one year after the outbreak of Coronavirus Disease 2019 (COVID-19), it remains largely unknown why, while most infected subjects develop an asymptomatic or mild disease course, a small proportion of patients progress towards respiratory failure, requiring invasive mechanical ventilation, and some die [1,2]. Patients experiencing severe COVID-19 are older and present comorbidities such as hypertension, obesity or diabetes [3,4]. Higher C-reactive protein, neutrophil-to-lymphocyte ratio, lactate dehydrogenase and interleukin (IL)-6 are normally found in severe COVID-19 and predict fatal outcome [5,6,7,8]. 

Important disturbances in the antiviral immune response characterize the severe forms of acute COVID-19, as SARS-CoV-2 evades the innate immunity by impairing the interferon (IFN) type I and III responses, and causes hyperinflammation with increased production of cytokines and chemokines [9,10,11]. COVID-19 patients may present important alterations in the structure and function of the monocyte compartment which could largely contribute to the observed immune dysregulation. In peripheral blood, acute COVID-19 patients show a decrease of non-classical (CD14−CD16+) and an increase of intermediate (CD14+CD16+) monocytes [12,13,14]. Isolated circulating monocytes from acute COVID-19 patients showed a sustained production of IL-6 and tumor necrosis factor (TNF)-α [15]. Monocytes show downregulation of surface HLA-DR expression, which is more pronounced in severe cases [10,12,15]. In bronchoalveolar lavage (BAL) and bronchoscopy secretions samples, there is accumulation of macrophages which derive from circulating monocytes [14,16,17] and produce high amounts of cytokines [18]. However, many of these evidence come from reduced numbers of patients belonging to cohorts of different severity and therefore different time of disease evolution, once they have received immunomodulatory therapies. Data obtained at emergency room (ER) arrival, before any therapeutic intervention, would help to understand the role of monocytes in the pathophysiology of COVID-19 and how they can impact the course of the disease. 

A considerable proportion of subjects who overcame COVID-19 suffers persistent and prolonged symptoms, a condition named long-COVID or post-acute COVID-19 syndrome [19,20,21]. Post-acute COVID-19 sequelae last beyond 4 weeks after the onset of the symptoms and may persist for more than 6 months. Persistent symptoms have been reported to affect from 30% to 80% of patients in different surveys [22]. Most common complaints include asthenia, dyspnea, anosmia, myalgia, arthralgia, anxiety or sleep disorders, in some cases accompanied by alterations in functional respiratory tests and lung or cardiac image [21,22]. It is suspected that long-lasting inflammation and immune dysregulation may underlie the symptoms and complications observed, and research studies searching for predictive immunologic biomarkers of long COVID-19 are ongoing [21,23]. An investigation of the immune system of patients suffering post-acute COVID-19 would be key to better understanding it and offering therapies.

Here we focused on studying circulating monocytes from fresh blood samples in confirmed COVID-19 patients at ER arrival, prior to receiving any treatment (acute COVID-19), in subjects who overcame COVID-19 six months ago (post-COVID-19) and in healthy controls (HC). In acute COVID-19 patient monocytes, we identified predominant transcriptional anti-inflammatory and senescence-related programs as well as structural changes in chromatin with comparatively increased accessibility. Some of these transcriptomic and epigenetic features were still visible six months later in post-COVID-19 monocytes. The early detected dysregulated monocyte compartment in acute COVID-19 included a particular subset with poor expression of surface markers and low IRF1 gene transcription. This monocyte subset defined acute COVID-19 patients with impaired adaptive immune response and significantly higher risk of entering intensive care unit (ICU) or dying.

## 2. Materials and Methods

### 2.1. Patients

We collected blood from 131 RT-PCR positive, acute-phase COVID-19 patients (acute COVID-19), who attended the University Hospital 12 de Octubre ER with SARS-CoV-2 infection, from 24 August to 5 November, 2020. In addition, we prospectively collected blood from acute COVID-19 patients at several times during their hospital stay. We also obtained samples from 52 patients who had overcome RT-PCR-confirmed COVID-19 six months before (Post-COVID-19). A total of 45 healthy controls (HC) were also included. All recruited subjects were unvaccinated against SARS-CoV-2. The institutional Clinical Research Ethics Committee approved the study protocol (reference no. 20/167) on 9 June 2020. 

### 2.2. PBMC Isolation and Flow Cytometry

Peripheral blood mononuclear cells (PBMC) were obtained from K2EDTA blood samples. Tubes were centrifuged and plasma was collected, aliquoted and stored at −20 °C. Cells were diluted with phosphate-buffered saline (PBS), PBMC were isolated by Ficoll density gradient and directly used for flow cytometry (FC) analysis and monocyte isolation.

Monocyte phenotype was analyzed by FC (FACS Canto II, BD Bioscience, San Jose, CA, USA). PBMC were stained with anti-CD3-V450, anti-CD19-V450, anti-CD56-V450, anti-CD14-FITC (BD Bioscience, San Jose, CA, USA), anti-CD33-PE-Cy7 (eBioscience, San Diego, CA, USA), anti-HLA-DR-APC, anti-CD16-V500, anti-CCR2-PE, anti-CCR5-APC-Cy7 and anti-CD86-PerCP-Cy5.5 (BD Bioscience, San Jose, CA, USA). Classical monocytes were defined as CD14+CD16−, intermediate monocytes as CD14+CD16+ and non-classical monocytes as CD14−CD16+. FC data were analyzed in FlowJo V10.

### 2.3. Monocyte Isolation and Stimulation

For ex vivo stimulation and cytokine production, monocytes were isolated from PBMC (EasySep Human monocytes enrichment kit w/o CD16, StemCell, Vancouver, Canada), according to manufacturer instructions. Purified monocyte fraction was stained with anti-CD14-FITC (BD Bioscience, San Jose, CA, USA) and anti-CD11b-APC-Cy7 (Beckman Coulter, Brea, CA, USA) and purity was evaluated via FC. Purified monocyte fraction was plated in 96-wells flat bottom plate (10^5^ monocytes/well) and incubated for 1 h in RPMI1640 supplemented with 10% fetal bovine serum (FBS), 1% penicillin-streptomycin, 1% L-glutamine and 1% sodium-pyruvate at 37 °C. After that, culture media was removed, and cells were washed with warmed PBS as extra of purification. New warmed culture media was combined with 10 ng/mL of LPS and monocytes were incubated overnight. Supernatant was collected and stored at −20 °C.

For RNA extraction, monocytes were isolated from PBMC (CD14 microbeads, MiltenyiBiotec, Bergisch Gladbach, Germany), according to manufacturer instructions. Purified monocytes were stained with anti-CD14-FITC (BD Bioscience, San Jose, CA, USA) and anti-CD11b-APC-Cy7 (BC), and purity was evaluated via FC. Purified monocytes (purity > 90%) were used to extract RNA.

### 2.4. Multiplex Detection of Cytokines 

Frozen plasma and supernatant samples were thawed and centrifuged. In plasma samples, 20 cytokines were detected using two multiplex detection kits (Human Cytokine/growth factor panel: G-CSF, IFN-α2, IL-1 α, IL-1Ra, CXCL10, CXCL9, CCL2 and VEGF-A; and high sensitivity panel: GM-CSF, IFN-γ, IL-10, IL-17A, IL-1β, IL-2, IL-21, IL-6, IL-7, IL-8, CCL3 and TNF-α; Millipore, Burlington, MA, USA) according to manufacturer instructions. In supernatant samples, 11 cytokines were detected using Human Cytokine/growth factor panel: IL-6, TNF-α, IL-10, IL-1β, GM-CSF, IL-18, IL-8, CCL3, IL-1Ra, CCL2 and CXCL10, according to manufacturer instructions (ThermoFisher Scientific, Waltham, MA, USA). Plates were analyzed by Luminex 100 (One Lambda, ThermoFisher Scientific, Waltham, MA, USA).

### 2.5. T Cell Proliferation Assay

CD14-negative cells were CFSE labeled, plated in a 96-well flat bottom plates (5 × 10^4^ cells/well), stimulated with coated anti-CD3 and soluble anti-CD28 and cultured in RPMI1640 supplemented with 10% fetal bovine serum (FBS), 1% penicillin-streptomycin, 1% L-glutamine and 1% sodium-pyruvate at 37 °C. After 4 days, cells were harvested and stained with antiCD4-V500 (Beckman Coulter, Brea, CA, USA) and antiCD8-PE-Cy7 (BD Bioscience, San Jose, CA, USA) and analyzed by FC. Percentage of proliferation was calculated as percentage of net CSFE dilution.

### 2.6. Fluorospot Assay

PBMCs were seeded at 3 × 10^5^ cells/well in IFN-γ IL-2 FluoroSpotTM plates (MabTech, Nacka Strand, Sweden) with cell culture medium containing RPMI, 1% L-glutamine, 1% penicillin/streptomycin, 10% fetal bovine serum and anti-CD28 mAb (1 µg/mL). Test wells were performed in duplicate and supplemented with 15-mer overlapping peptides covering the S1 domain of the S glycoprotein (166 peptides) (SARS-CoV-2 S1 scanning pool, MabTech, Nacka Strand, Sweden), the N protein (102 peptides) (PepMixTM SARS-CoV-2 [NCAP], JPT) and the M protein (53 peptides) (PepMixTM SARS-CoV-2 [VME1], JPT) at a final concentration of 1 µg/mL. Negative control wells lacked peptides, and positive control wells included anti-CD3 mAb (MabTech). Assays were incubated for 16–18 h at 37 °C. Spots were counted using an automated IRISTM FluoroSpot Reader System (MabTech, Nacka Strand, Sweden). To quantify antigen-specific responses, spots of the negative control wells were subtracted from the mean spots test wells, and the results were expressed as IFN-γ-or IL-2-producing spot forming units (SFUs) per 10^6^ PBMCs. 

### 2.7. RNA Extraction and Nanostring Assay

Total RNA extraction from monocytes was preformed using the RNeasy Plus kit (Qiagen, Hilden, Germany) according to manufacturer instructions. RNA concentration was determined using NanoDrop 1000 spectrophotometer (ThermoFisher Scientific, Waltham, MA, USA) and Qubit^®^ RNA HS Assay (ThermoFisher Scientific, Waltham, MA, USA). RNA quality and integrity were assessed using the Agilent RNA 6000 Nano kit with the Agilent 2100 Bioanalyzer instrument (Agilent Technologies Inc., Santa Clara, CA, USA). RNA samples were used when the concentration was ≥20 ng/μL and RNA integrity Number (RIN) was >9.100 ng of RNA was hybridized to target-specific probes and controls in a single tube for 20 h at 65 °C. Target-probe complexes were purified and immobilized on the nCounter prep station. Counts for each target RNA were acquired using the nCounter detection analyzer (NanoString, Seattle, WA, USA). Finally, nSolver software (NanoString, Seattle, WA, USA) was used for normalization using housekeeping genes and for paired comparisons of gene expression. Volcano plots and heatmaps were performed using ggplot2 and ComplexHeatmap R packages.

### 2.8. ATAC Sequencing

For ATAC-Seq, 50,000 monocytes from HC, acute and post-COVID-19 patients were isolated and cryopreserved in 10% DMSO + 90% SF (*n* = 3 for each condition). After being rapidly unfrozen, cells were counted, lysed (10 mM PIPES pH 6.8, 100 mM NaCl, 300 mM sucrose, 3 mM MgCl2, 0.1% TritonX-100 for 5 min at 4 °C and centrifuged at 800× *g*. Supernatant was discarded, and nuclei were resuspended in transposase reaction buffer with transposase5 (1 µL Tn5/10.000 cells) and incubated for 30 min at 37 °C. Later, SDS at a final concentration of 0.1% was added to stop the reaction (30 min, 40 °C) and DNA was isolated with KAPA Pure beads (SPRI beads). Briefly, lysates were incubated with SPRI beads (2×), placed on the magnet, washed twice with 70% ethanol, and air-dried. DNA was eluted with elution buffer (EB) and its concentration measured with Qubit kit (Invitrogen, Waltham, MA, USA). For DNA library generation, qPCR with i5 and i7 primers (NEXTERA barcoding) was performed (98 °C 2 min, 98 °C 20 s, 63 °C 30 s, 72 °C 1 min, 4 °C ∞; 14 cycles). Then, a size cut-off was conducted. First, amplified DNA was incubated with 0.5× concentration of SPRI beads. Then, beads were discarded, and srnatant was incubated with 1.8× concentration of SPRI beads. Beads were washed twice with 70% ethanol, eluted with EB, and measured with Qubit. Finally, Agilent HS Bioanalyzer was used to assess peak sizes and percentages.

Libraries were sequenced on a HiSeq 4000 (Illumina, San Diego, CA, USA) to generate at least 30 million paired reads. Sequencing outputs were processed with RTA v1.18.66.3. FastQ files for each sample were obtained using bcl2fastq v2.20.0.422 software (Illumina, San Diego, CA, USA). Read quality was assessed using FastQC. Sequencing reads were trimmed for Illumina adapter and transposase sequences with cutadapt, aligned to the human reference genome (GRCh38) with bowtie and exclude PCR duplicates with samtools MarkDuplicates tool.

Peaks were identified with HOMER2 in region mode, with a local size of 50,000 bp, a peak expected size of 150 bp, minimum non-unified distance of 1000 bp, minimum coverage of 5 reads and a region resolution of 6. For each condition, peaks that were detected in all replicates were taken into account. For the final peak file, all peaks detected in any of the 3 conditions were used. Peaks were also annotated with HOMER2.

Bigwig files for visualization were created with Deeptools and Wiggletools. Tracks were represented in IGV browser. Final aggregated figures were represented with custom python scripts available under request.

### 2.9. Statistics

Continuous numeric variables are represented as median and interquartile range (IQR) and were compared by using Mann–Whitney U test. Categorical variables are represented as N and percentage and were compared by using Fisher exact test or Chi-square test when appropriate. Numerical variables were plotted as violin plot. Correlation was evaluated with linear regression and the Spearman rank coefficient and were represented as correlation plots using Hmisc and corrplot R packages. In correlation plots, the size of the square is proportional to Spearman’s coefficient, the color indicates if the correlation is positive or negative and asterisks indicate that this correlation is significant. Principal component analysis (PCA) of log-transformed cytokine levels and biplot were performed using factoMineR and factorextra R packages. 

For k-means clustering, MFI of monocytes markers were log10-transformed and center scaled. Number of clusters was identified by using silhouette’s index implemented in NbClust R package. ComplexHeatmap R package was used for representing heatmap. Relationship between cluster classification and ICU or death was performed by logistic regression and odds ratios were obtained. Kaplan-Meier analysis was performed and represented using the survival and ggpubr R packages, curves were compared using Log rank test. All the analysis was performed in R v 4.0.3. Differences were considered statistically significant when *p*-value < 0.05.

## 3. Results

### 3.1. Study Patients 

We enrolled 131 consecutive patients who attended the ER and were hospitalized with acute COVID-19. All patients had a positive real time polymerase chain reaction (RT-PCR) for SARS-CoV-2 (acute COVID-19 patients). In addition, we obtained samples from another 52 patients 6 months after hospitalization because of positive RT-PCR, SARS-CoV-2 infection (post-COVID-19) and from 45 age- and gender-matched healthy controls (HC). Acute COVID-19 and post-COVID-19 patient characteristics are detailed in Appendix A.

### 3.2. Phenotype and Function in Acute and Post-COVID-19 Circulating Monocytes

We analyzed circulating monocyte subsets in HC, acute and post-COVID-19 patients (Figure 1A). At hospital arrival, acute COVID-19 patients had a significant increase of classical (84.5% vs. 77.7% in HC, *p* < 0.0001) and intermediate (11.5% vs. 5.7% in HC, *p* < 0.0001) and significant reduction of non-classical (1.0% vs. 6.8% in HC, *p* < 0.0001) monocytes, as previously reported [10,12,13,14]. In post-COVID-19 patients, intermediate monocytes were lower than in acute patients but still significantly increased when compared to HC (7.0% vs. 5.7%, *p* = 0.03), while classical and non-classical subsets were similar to those in HC (Figure 1A and Appendix A). 

Next, we analyzed monocytes according to their expression of surface molecules. The principal component analysis (PCA) showed that there were overall differences among HC, acute and post-COVID-19 subjects (Figure 1B). When compared to HC, monocytes from acute COVID-19 patients had higher size (FSC) and granularity (SSC), higher expression of CD11b, CD16 and CD33, similar expression of CCR2, CCR5 and CD86, and a marked downregulation of HLA-DR, as previously reported [12,15,24] (Figure 1C). On the other hand, monocytes from post-COVID-19 patients still showed higher size than HC monocytes but their granularity was significantly reduced by comparison to acute COVID-19 and HC monocytes. When compared to acute COVID-19 monocytes, the expression of CD16 and CD33 in post-COVID-19 monocytes was downregulated, equaling the level found in HC monocytes in the case of CD16. Interestingly, post-COVID-19 monocytes showed the highest expression of CCR2, CCR5, CD86 and HLA-DR, and the lowest of CD11b (Figure 1C). 

The function of monocytes was interrogated by measuring the production of cytokines after in vitro stimulation of isolated monocytes with lipopolysaccharide (LPS). In comparison to HC, monocytes from acute COVID-19 patients secreted higher amounts of the pro-inflammatory cytokines GM-CSF, TNF-α, IL-1b, IL-6, IL-8, IL18 and CCL2 and CCL3 (the macrophage inflammatory protein-1α, MIP-1α) and lower amounts of CXCL10 (interferon gamma-induced protein IP-10) and the anti-inflammatory cytokines IL1-Ra and IL-10. This observation confirms the hyperresponsiveness status of monocytes in acute COVID-19, as opposed to the monocyte paralysis observed in sepsis patients [25]. In post-COVID-19 subjects, the cytokine production of monocytes was similar to that of HC, except for CXCL10, which was synthesized significantly less than in HC (Figure 1D).

In summary, circulating monocytes from acute COVID-19 patients, as examined at hospital arrival, were mostly classical CD14+CD16− monocytes with significant downregulation of HLA-DR and increased capacity to secrete pro-inflammatory cytokines. Conversely, post-COVID-19 subjects had normal proportions of monocyte subsets which had returned to a normal basal resting state and expressed the highest CCR2, CCR5, CD86 and HLA-DR levels on their surface.

### 3.3. Monocyte Relationship with Cytokine Environment and SARS-CoV-2-Specific T Cell Response in Acute and Post-COVID-19 Patients 

Acute COVID-19 patients differed from HC and post-COVID-19 based on 20 plasma inflammation and immunity mediators (Figure 2A). IL-6, CXCL10, CXCL9, CCL2, IL1-Ra, IL-10, G-CSF and TNF-α were the cytokines with the highest contribution to dimension 2 in the PCA (Figure 2B) and were significantly increased in acute COVID-19 patients in comparison to HC (Appendix A). In contrast, the highest contributors to dimension 1 were IL-21, IL-2, IL-1b, IL-17A, GM-CSF, IFN-γ, IL-7 and CCL3 (Figure 2B), and these cytokines were significantly decreased in acute COVID-19 patients in comparison to HC (Appendix A). In particular, IL-6 emerged as the most increased circulating cytokine in acute COVID-19, followed by G-CSF and CXCL10 (blue dots in Figure 2C). Six months after recovery, these soluble mediator levels had returned to normal (Appendix A); however, G-CSF and IL-6 were still elevated in post-COVID-19 patients compared to HC (grey squares in Figure 2C). In a correlation analysis between the expression of molecules on the monocyte surface and circulating cytokine levels, HLA-DR showed the highest number of significant correlations. HLA-DR expression on monocytes showed a significant inverse correlation with the three highest cytokines in COVID-19 patients: IL-6 (r = −0.36, *p* < 0.001), G-CSF (r = −0.22, *p* = 0.02) and CXCL10 (r = −0.22, *p* = 0.01); and also with CCL2 (r = −0.28, *p* = 0.001), together with a direct correlation with IFN-γ (r = 0.22, *p* = 0.04) (Figure 2D). 

Lymphopenia (<1.2 × 10^3^ lymphocytes per µL) was observed in 72.5% of our acute COVID-19 patients, and it was accompanied by a reduced T cell proliferation capacity (Figure 2E) which was restored in post-COVID-19 subjects. Expression of HLA-DR and CCR5 in acute COVID-19 monocytes positively correlated with CD4 and CD8 T cell proliferation, reaching significance in the case of CD4 T lymphocytes (Appendix A). Few acute COVID-19 patients showed positive SARS-CoV-2-specific T cell responses at admission. On the contrary, most post-COVID-19 patients showed positive, strong specific T cell responses (Figure 2F). No significant correlations were found between specific T cell responses and monocyte subsets; however, a positive significant correlation was found between IFN-γ S1-specific T cell response and several surface expression molecules in monocytes which included HLA-DR and CD86, both in acute and post-COVID-19 patients (Figure 2G). 

Altogether, these results suggested that in early acute COVID-19 patients, monocytes with HLA-DR downregulation were associated with a pro-inflammatory cytokine environment and limited SARS-CoV-2-specific T cell response. In turn, normalization of pro-inflammatory cytokines in concomitance with augmentation of HLA-DR expression in monocytes and capacity to mount an antiviral cellular response were characteristic of post-COVID-19.

### 3.4. Inflammation-Resolution Gene Expression in Monocytes from Acute and Post-COVID-19 Patients 

We assessed the expression of 255 inflammation-related genes in freshly isolated monocytes from HC (N = 7), early acute (N = 18) and post-COVID-19 patients (N = 8). Monocyte gene expression clearly separated HC, acute and post-COVID-19 patients (Figure 3A). In the pairwise differential gene expression analysis between HC and acute COVID-19, we observed 24 up- and 18 downregulated genes in monocytes from acute COVID-19 patients in comparison to HC (Figure 3B). Interestingly, monocytes from acute COVID-19 showed a predominant upregulation of anti-inflammatory genes, such as IL-10, BCL6, MAFF, MAFG and AREG, together with downregulation of pro-inflammatory cytokines such as TNF-α and CCL3 and downregulation of the antigen presentation molecule HLA-DRA (in agreement with flow cytometry data) or CD4. These findings suggest that circulating monocytes from acute COVID-19 patients may be contributing to inflammation resolution and to tissue repair. Albeit to a lesser extent, IL-1RAP, OASL and MX1 genes, which belong to the “senescence associated secretory phenotype” (SASP) [26], were also found significantly upregulated in acute COVID-19 vs. HC monocytes.

Six months after COVID-19 recovery, monocytes showed a different gene expression profile than in acute COVID-19 patients although not yet identical to that of HC monocytes. Post-COVID-19 monocytes had 9 up- and 17 downregulated genes in comparison to HC (Figure 3C). Among the upregulated genes in post-COVID-19 monocytes, we still found the acute-COVID-19 upregulated BCL6, AREG and IL-10.

Gene expression in monocytes from acute and post-COVID-19 patients revealed 28 differentially expressed genes (DEG), with 9 up- and 19 downregulated genes in post-COVID-19 patients (Figure 3D). Post-COVID-19 monocytes showed “back-to-normal” expression features such as the downregulation of IL-10 and the upregulation of CD4 and HLA-DRA, which was in agreement with the flow cytometry data showing an increase of HLA-DR expression in monocyte surface after recovery (Figure 1C).

In summary, the transcriptome of circulating monocytes from acute COVID-19 patients had a predominant expression of genes with inflammation-resolution and tissue-protection functions, and this signature was still prevalent in post-COVID-19 monocytes.

### 3.5. Changes in Chromatin Accessibility of Monocytes from HC, Acute and Post-COVID-19 Patients

To further understand the differential features observed in monocyte phenotype, activation and transcriptomic profiling from HC, acute and post-COVID-19 patients, the genome-wide accessibility of monocyte chromatin was analyzed in three patients from each cohort by ATAC-Seq. Information of chromatin structure was obtained from a total of 16,553 sites. Each one of the three cohorts showed several exclusive open chromatin sites: 301 in HC, 3830 in acute COVID-19 and 1145 in post-COVID-19 monocytes (Figure 4A). Thus, the distribution of transposase-accessible sites across the whole genome showed a relative increment of open chromatin sites in acute COVID-19 and it was the lowest in HC monocytes, while post-COVID-19 monocytes showed an intermediate open chromatin profile (Figure 4B). 

Some open regions found, such as those in MAFF and CD4, corroborated the transcriptomic results. The MAFF-promoter region showed increased accessibility in acute COVID-19 monocytes in comparison to monocytes from post-COVID-19 and HC subjects (Figure 4C), whereas CD4 chromatin opening was reduced in monocytes from the acute disease, intermediate in post-COVID-19 monocytes and maximum in HC (Figure 4D). In addition to those coinciding with transcriptomic data, many differentially open sites were detected in other genes. Of note, we found that acute- and post-COVID-19 monocytes had a higher accessibility in the promoter region of lamin-B receptor (LBR) (Figure 4E), a gene related to protection against cellular senescence [27]. In summary, we found an association of SARS-CoV-2 infection with changes in the chromatin accessibility pattern of circulating monocytes, some of which were still maintained six months after hospitalization.

### 3.6. Analysis of Monocyte Markers at Hospital Admission Allows to Identify Acute COVID-19 Patients at High Risk of ICU Requirement and Death 

In the PCA analysis based on the surface expression of molecules in monocytes, we observed an important dispersion of acute COVID-19 patients along dimension 1 (Figure 1B). The k-means clustering of the data revealed two types of patients, which were named cluster A and cluster B (Figure 5A). When compared to cluster B, cluster A patients showed significantly more classical and less intermediate monocytes (Figure 5B). Cluster A monocytes presented lower expression of CD16, CCR2, CCR5, CD86, HLA-DR and CD33, only CD11b expression was similar between monocytes from both clusters (Figure 5C). 

In cluster A we observed a significant accumulation of patients who were subsequently admitted to ICU (23.3% vs. 9.7% in cluster B, *p* = 0.05) or died (23.3% vs. 5.6% in cluster B; *p* = 0.03) (Figure 5D and Appendix A). Acute COVID-19 patients classified as cluster A according to monocyte markers at ER arrival had 2.78 [95% CI 1.07–7.85, *p* < 0.05] and 3.05 [1.19–8.55, *p* < 0.05] higher risk of ICU requirement and death, respectively. The Kaplan–Meier analysis showed not only increased mortality in cluster A patients (*p* = 0.034), but also that deaths occurred significantly sooner after hospitalization than in cluster B (4 (3–15) days vs. 21 (19–29) days, *p* = 0.01) (Figure 5E,F). Remarkably, known factors associated with a critical course of the infection, such as age, sex, comorbidities, ARDS, ratio of peripheral blood oxygen saturation to fraction of inspired oxygen (SpO2/FiO2), monocytes, platelets, neutrophil-to-lymphocyte ratio, LDH, albumin, AST, ALT, fibrinogen, D-dimer, viral load (Ct) [28,29,30,31], days of symptoms and treatment received were not different between clusters. The only parameters that differed were IL-6 and C-reactive protein, which were both significantly increased in cluster A (Appendix A), suggesting that cluster A includes a group of patients with higher early inflammation. Consistent with the reported lack of association between in vitro production of monocytes-derived cytokines and severity [32] there was no difference in the production of cytokines by cluster A or B monocytes after in vitro stimulation with LPS (Appendix A). However, the evaluation of a broad panel of plasma cytokines showed overall differences between clusters A and B (Appendix A). In support of cluster A containing a group of patients with superior early inflammation, higher levels of most circulating cytokines were detected in cluster A vs. B, reaching the significance in the case of IL-1Ra, IL-6, IL-8 and CCL3 (Appendix A). 

We observed that the proliferative capacity of polyclonal CD4 T cells was similar in clusters A and B, while that of CD8 T cells was higher in the COVID-19 patients from cluster B (Figure 5G). In addition, cluster B patients showed a trend towards superior spike (S1)-, nucleocapsid (N)- and membrane (M)-specific T cell responses as enumerated by IFN-γ or IL-2 spots forming units (SFU)/10^6^ PBMCs (Figure 5H), and cluster B accumulated significantly more patients with ≥15 S1-specific IFN-γ SFU/10^6^ PBMC than cluster A (Figure 5I), suggesting that cluster B patients had a more robust SARS-CoV-2-specific cellular immune response as early as a week after symptom onset. 

Monocytes and SARS-CoV-2-specific T cell response were longitudinally analyzed in a subgroup of acute COVID-19 patients (*n* = 78, 38 belonging to cluster A and 40 belonging to cluster B). The expression of most surface markers in monocytes gradually increased from week 1 to week 2 of hospitalization in patients who survived (Appendix A). HLA-DR augmentation was particularly significant in surviving patients and associated with a growing T cell response, whereas a progressive drop in HLA-DR monocyte surface expression together with a T cell unresponsiveness were observed in patients who subsequently died (Figure 5J,K and Appendix A). 

The inflammation-related, 255-gene transcriptomic profile of isolated monocytes from cluster A and B patients, during the first ER evaluation, revealed only one significant DEG, coding for the interferon regulatory transcription factor 1 (IRF1) (Figure 5L). IRF1 was downregulated in cluster A in comparison to cluster B monocytes (Figure 5M), suggesting that the superior degree of severity of cluster A patients could be related to a poor interferon type I response.

## 4. Discussion

We first set out to characterize the changes of circulating monocytes at the early phase of acute SARS-CoV-2 infection, and how these monocytes returned to homeostasis after recovery. We confirmed that in comparison to HC, in early acute COVID-19 non-classical monocytes were reduced while classical and intermediate monocytes predominated and showed downregulation of HLA-DR surface expression [10,12,13,14,15,33,34]. In patients who had overcome the disease 6 months previously, the monocyte distribution resembled more closely that of HC and monocytes showed a high expression of HLA-DR and CD86, which suggested a recovery of antigen presentation and co-stimulation capacity. Differently from the immunoparalysis observed in sepsis [25,35,36], circulating monocytes at the early phase of acute COVID-19 showed an increased basal activation as demonstrated by the augmented production of pro-inflammatory and reduction of anti-inflammatory cytokines after in vitro stimulation. This observation is consistent with a M1-polarization of the immune response and suggests that circulating monocytes may contribute to the global inflammatory status and exacerbate the cytokine storm in severe COVID-19 cases [37,38,39]. 

When analyzing the soluble cytokine environment, IL-6, G-CSF and CXCL10 were the most abundant in early phase, acute COVID-10 patients. HLA-DR expression in monocytes showed an inverse correlation with these three cytokines while it directly correlated with IFN-γ, as well as with polyclonal lymphocyte proliferation and SARS-CoV-2-specific cellular immune response. It will be relevant in future studies to elucidate if the changes detected in monocytes during acute COVID-19 are related with the observed limited T cell proliferation and through which mechanisms. Interestingly, G-CSF, IL-1β and IL-6 were still elevated in post-COVID-19 compared to HC. The high level of IL-1β in post-COVID vs. HC was particularly remarkable, since patients with acute COVID-19 had relative lower levels of this cytokine than post-COVID-19 and HC subjects. It would be of interest to investigate if the increase of inflammatory cytokines in post-COVID-19 is accompanied by persistent symptoms. This remains unexplored in our cohort, and is a limitation of the current work.

In contrast with the M1 features described above, a predominance of M2, anti-inflammatory program gene expression was observed in the transcriptome of circulating monocytes from acute COVID-19. This program was mostly represented by the significant upregulation of BCL6, MAFF, MAFG, AREG and IL-10. BCL6 has been described as a negative regulator of monocyte/macrophage activation and proliferation, and a repressor of monocyte-derived chemokines and IL-6 transcription [40,41,42]. MAFF and MAFG belong to the family of Maf transcription factors. Some Maf family members are involved in the maintenance of a M2-like program in macrophages, favoring homeostasis and tissue repair [43,44]. In a mouse model of LPS-induced acute lung injury it was observed that the classically activated alveolar macrophages also protected the lung tissue through production of AREG (amphiregulin) [45]. Data from a single-cell transcriptomic analysis of PBMC in severe COVID-19 patients showed a significant enhancement of MAFF and AREG expression in monocytes [46]. Altogether, our findings of augmented AREG or IL-10 transcription together with low HLA-DR and CD86 surface expression may represent a population of immature monocytes with suppressive properties, mostly augmented in acute COVID-19 patients with more lymphopenia, reduced specific-IFN-γ response and poor prognosis. Downregulation of HLA-DR and CD86 in cell membrane and production of AREG are hallmarks of monocytic-myeloid derived suppressor cells which have been found expanded in circulation in severe and non-survivor COVID-19 patients [33,47,48].

Interestingly, although with a lesser intensity, the upregulation of AREG, MAFG, MAFK, BCL6 and IL10 still persisted in monocytes of post-COVID-19 subjects. If upregulation of those genes in post-COVID-19 monocytes is related with long-COVID persistent symptoms remains unknown. On the other hand it has been shown that the in vitro infection of monocytes and monocyte-derived-macrophages with SARS-CoV-2 is associated with the induction of a M2-type transcriptional program [49]. Based on this result, we cannot discard the notion that the anti-inflammatory transcriptomic profile in acute COVID-19 monocytes observed here is derived from SARS-CoV-2-infected circulating monocytes, which could still remain in some post-COVID-19 patients. Monocytes and macrophages express ACE2 and TMPRSS2 surface molecules needed to bind and internalize SARS-CoV-2, viral RNA and nucleocapsid proteins have been detected in CD169+, splenic marginal zone macrophages, and SARS-CoV-2 RNA has been read in single-cell RNA analysis from macrophages [50,51,52]. Thus, an inflammation-resolution expression program characterizes the transcriptome of circulating monocytes during acute COVID-19 and may be related with the myeloid-driven immune suppression observed in other studies [32,53]. This program is still observed in convalescent phases of COVID-19.

In addition, a gradually closed chromatin accessibility profile was found in monocytes from acute-, post-COVID-19 and HC, suggesting that epigenetic changes are induced during the acute phase of COVID-19 which last several months after recovery. It has been shown that the long-lasting epigenetic changes observed in monocytes may correspond to a trained immunity mechanism caused by the capacity of different stimuli to reprogram myeloid progenitor cells [54]. Thus, our results suggest that SARS-CoV-2 infection could induce trained immunity.

The observation of concordant data between ATAC-Seq and transcriptomic studies is limited because of the different breadth between both analyses, since ATAC-Seq interrogated the whole genome whereas the expression analysis included 225 inflammation-related genes. Nevertheless, in particular cases such as MAFF and CD4, the structural profile in chromatin was concordant with the expression data. Of interest, the opening of the senescence-protector gene LBR [27] was superior in acute- and post-COVID-19 monocytes. LBR reduction has been related to diminished cellular cholesterol synthesis, alterations in the chromatin structure which limit the cell cycle progression and promotion of a SASP with upregulation of IL-6, IL-8 and CCL3 [55]. We found an SASP including IL1RAP, OASL and MX1 genes [26] significantly upregulated in acute COVID-19. Thus, the increased availability of LBR product may be interpreted as a compensatory mechanism against senescence in the circulating monocytes of COVID-19 patients.

Importantly, a poor expression of surface markers and low transcription of IRF1 gene in circulating monocyte allow us to define a cluster of acute COVID-19 patients at admission with impaired adaptive immune response and increased risk of requiring ICU or dying. Conversely, a monocyte program consisting of upregulation of IRF1 and a pro-inflammatory M1 profile, with abundant surface markers expression and antigen presentation ability, was linked to the capacity to mount a robust anti-SARS-CoV-2 specific cellular response and determined a benign disease course. The longitudinal measurements in acute COVID-19 patients showed that the progressive increase of HLA-DR expression in monocytes paralleled an increasing specific T cell response, and this was mostly observed in patients who survived, whereas patients who failed to augment HLA-DR expression in monocytes also failed to mount protective anti-SARS-CoV-2 cellular immune responses and died. These results agree with previous data which suggested a mortality predictive value of dual monocyte state and gene modules in severe COVID-19 [56].

Among the interrogated 255-gene panel including the interferon regulatory factors-1, -3, -5 and -7 in monocytes with low or high expression of surface markers, IRF1 was the only one differentially expressed. IRF1 was significantly augmented in monocytes with upregulated surface markers from patients later experiencing a benign course of the disease. Consistently, previous reports have shown that IRF1 expression in classical monocytes was lower in severe than in moderate COVID-19 patients [57]. IRF1 mediates anti-viral defense by multiple constitutive and inducible mechanisms, the later including pathways dependent from RIG-I-like, toll-like, IFN and TNF-α receptors [58]. While IRF1 expression is mostly dependent from type I IFNs [59], it may stimulate type I and II IFNs to suppress the replication of different RNA viruses [60], or it may stimulate IL-12 to facilitate the expression of inflammatory genes associated with a M1 phenotype [61]. These data suggest a close association between early, IRF1-dependent protective mechanisms from circulating monocytes and a less severe course of COVID-19.

## 5. Conclusions

In conclusion, the acute phase of COVID-19 is characterized by a dysregulated compartment of circulating monocytes, with deeper alterations in patients who will develop a more severe disease course. The early flow cytometry analysis of peripheral blood monocytes, before any therapeutic intervention, may provide predictive information on COVID-19 patient outcome. Acute COVID-19 monocytes express higher levels of inflammation-resolution and tissue homeostasis genes, together with a superior overall chromatin accessibility, and some of these features are still present in monocytes 6 months after recovering. In addition to shedding light on the monocyte disturbances underlying COVID-19 and their duration after acute infection, this study may be useful for the design of innate-immunity focused therapies. 

## Figures and Tables

**Figure 1 biomedicines-09-01253-f001:**
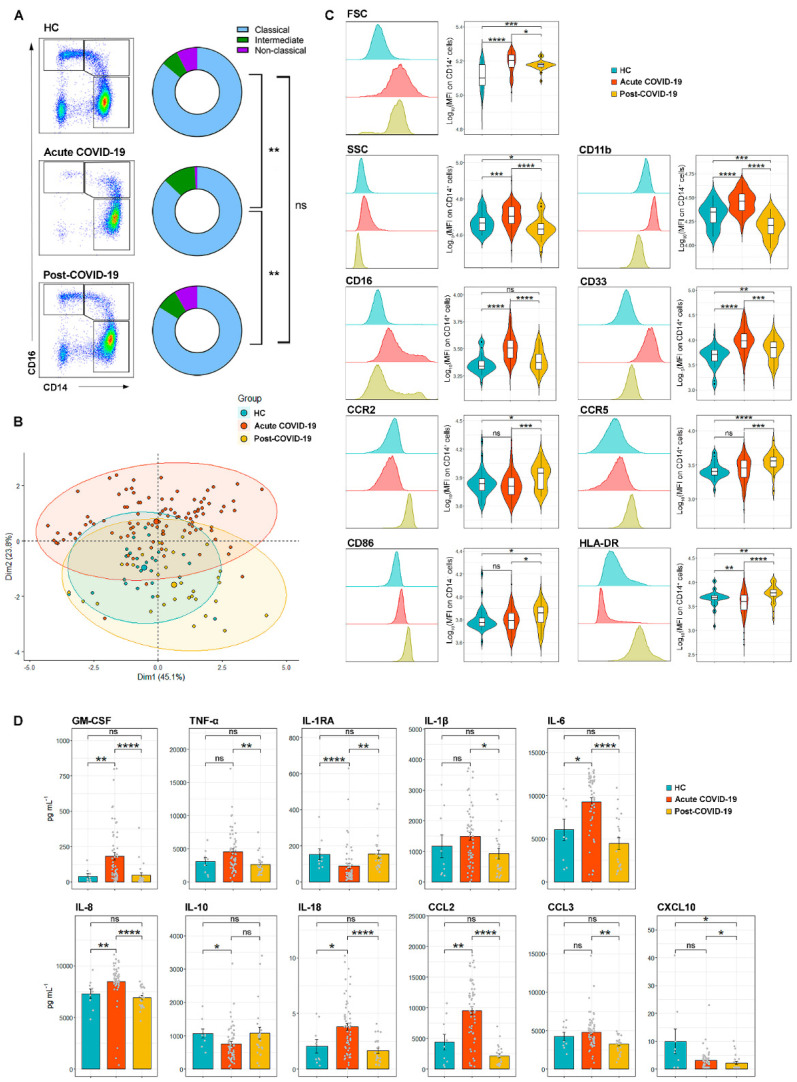
Circulating monocyte subsets, their surface molecule expression and their cytokine production after in vitro activation are altered in acute and post-COVID-19 patients. (**A**) Identification of classical (CD14+CD16-), intermediate (CD14+CD16+) and non-classical (CD14−CD16+) monocytes. In HC, classical monocytes represented 77.7%, intermediate monocytes 5.7% and non-classical monocytes 6.8%. In acute COVID-19 those populations were 84.5%, 11.5% and 1.0% respectively, and in post-COVID-19 they were 75.5%, 6.9% and 7.6% respectively. (**B**) Principal component analysis (PCA) of HC (*n*= 45), acute (*n* = 131) and post-COVID-19 patients (*n* = 52) according to monocyte surface marker expression. Individual patient values (small points), mean (big point) and confidence interval of the mean (shadows) are represented (**C**) Examples and comparison of FSC, SSC, CD11b, CD16, CD33, CCR2, CCR5, CD86 and HLA-DR expression in monocytes from HC, acute and post-COVID-19 patients. (**D**) Cytokines secreted by isolated, LPS-stimulated monocytes. In unstimulated monocyte wells all cytokines were below the detection threshold: GM-CSF < 2.6 pg/mL; TNF-α < 6.4 pg/mL; IL1-RA < 1.6 pg/mL; IL1β < 1.6 pg/mL; IL-6 < 0.64 pg/mL; IL-8 < 0.64 pg/mL; IL-10 < 2.6 pg/mL; IL-18 < 0.64 pg/mL; CCL2 < 3 pg/mL; CCL3 < 3 pg/mL; CXCL10 < 2.6 pg/mL.*, *p* < 0.05; **, *p* < 0.01; ***, *p* < 0.001; ****, *p* < 0.0001; ns, non-significant.

**Figure 2 biomedicines-09-01253-f002:**
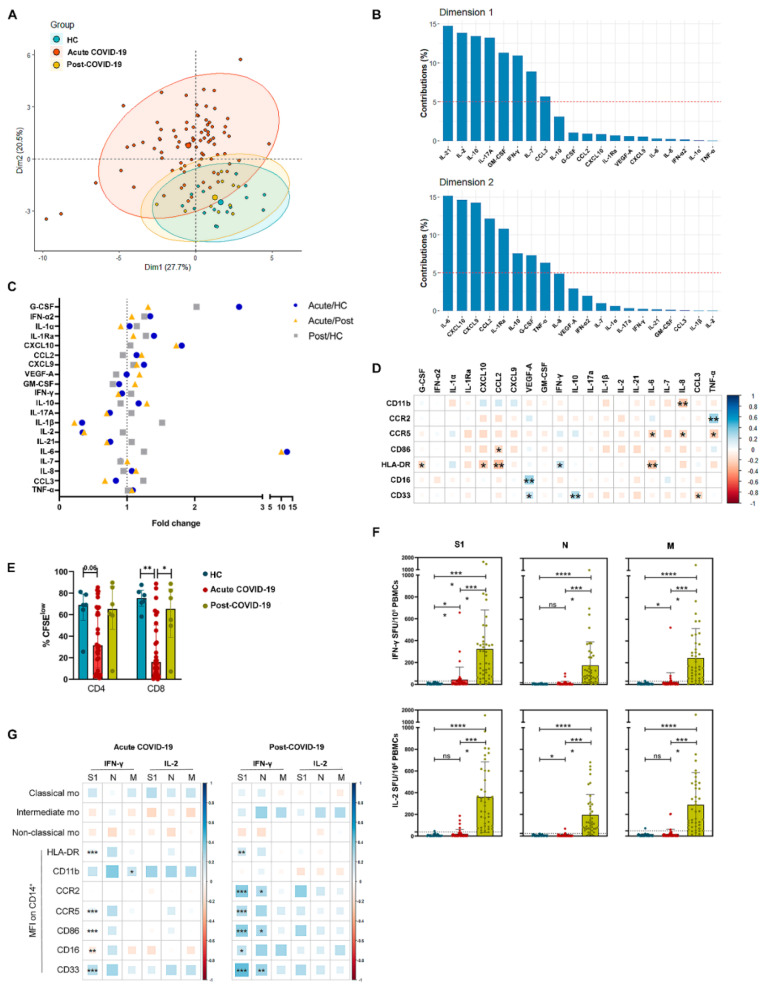
HLA-DR downregulation in monocytes correlates with higher plasma levels of IL-6, G-CSF, CXCL10, CCL2 and lower SARS-CoV-2-specific T cell response. (**A**) PCA of 20 cytokines in plasma from HC (*n* = 16), acute (*n* = 84) and post-COVID-19 (*n* = 10) patients. Individual patient values (small points), mean (big point) and confidence interval of the mean (shadows) are represented. (**B**) Contribution of each cytokine to PCA dimensions 1 and 2. Dashed line represents the mean of all cytokine contributions. (**C**) Fold changes of median log-transformed values of plasma cytokines analyzed in HC, acute and post-COVID-19 patients. Dotted line represents unchanged cytokine level. (**D**) Correlation analysis between plasma cytokine levels and monocyte surface expression of CD11b, CCR2, CCR5, CD86, HLA-DR, CD16 and CD33. (**E**) Proliferation of in vitro polyclonally stimulated CD4 and CD8 T cells from HC, acute COVID-19 and post-COVID-19. (**F**) SARS-CoV-2 S1, N and M specific cellular response in HC (*n* = 30), acute (*n* = 40) and post-COVID-19 (*n* = 41) patients. Dashed lines represent the established cut-off of positivity according to HC SFU means plus 3 standard deviations. (**G**) Correlation between specific T cell response and monocyte subsets and surface markers in acute and post-COVID-19 patients. Color of squares indicates direct (blue) or inverse (red) correlation. The size of the square is proportional to magnitude of correlation. *, *p* < 0.05; **, *p* < 0.01; ***, *p* < 0.001; ****, *p* < 0.0001; ns, non-significant.

**Figure 3 biomedicines-09-01253-f003:**
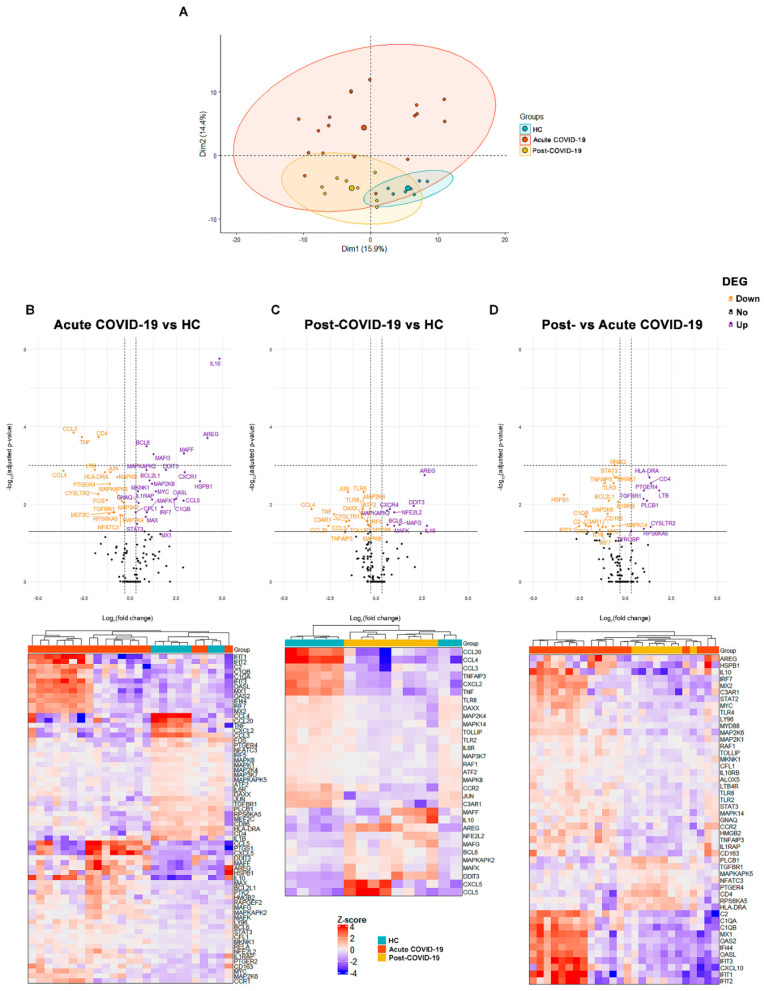
The transcriptomic profiling of circulating monocytes reveals an anti-inflammatory program in early acute COVID-19 still detectable in post-COVID-19 subjects. (**A**) PCA based on 255 analyzed genes clearly separated monocytes from HC, acute and post-COVID-19 patients. DEG of monocytes from (**B**) HC and acute COVID19, (**C**) HC and post-COVID19, (**D**) acute and post-COVID19. In (**B**–**D**), top panels depict volcano plots showing the genes with significant differences, horizontal solid line indicates a *p*-value < 0.05 and horizontal dashed line indicates a *p*-value < 0.001; vertical lines indicate log2 fold change <−0.6 and >0.6; bottom panels depict heatmaps showing the individual expression of significant DEG in each subject.

**Figure 4 biomedicines-09-01253-f004:**
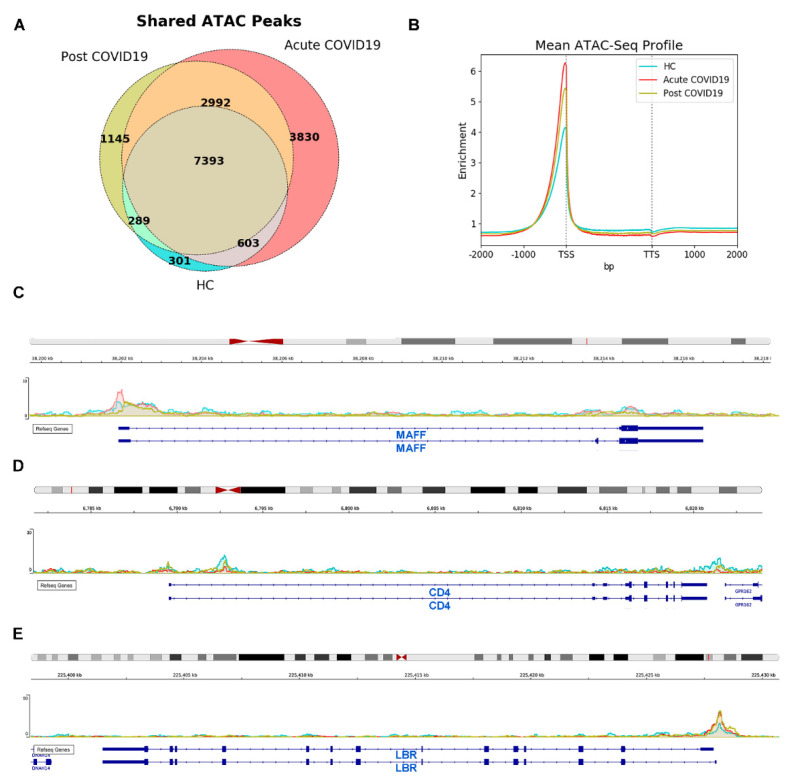
Open chromatin profile is higher in acute, intermediate in post-COVID-19 and lower in HC circulating monocytes. (**A**) Venn diagram of open sites found in HC, acute and post-COVID-19 monocytes. (**B**) Global enrichment sites in HC, acute and post-COVID-19 monocytes. (**C**) MAFF-promoter region had greater opening in acute COVID-19 monocytes. (**D**) CD4-promoter region had greater opening in HC monocytes. (**E**) LBR-promoter region had greater opening in acute and post-COVID-19 monocytes.

**Figure 5 biomedicines-09-01253-f005:**
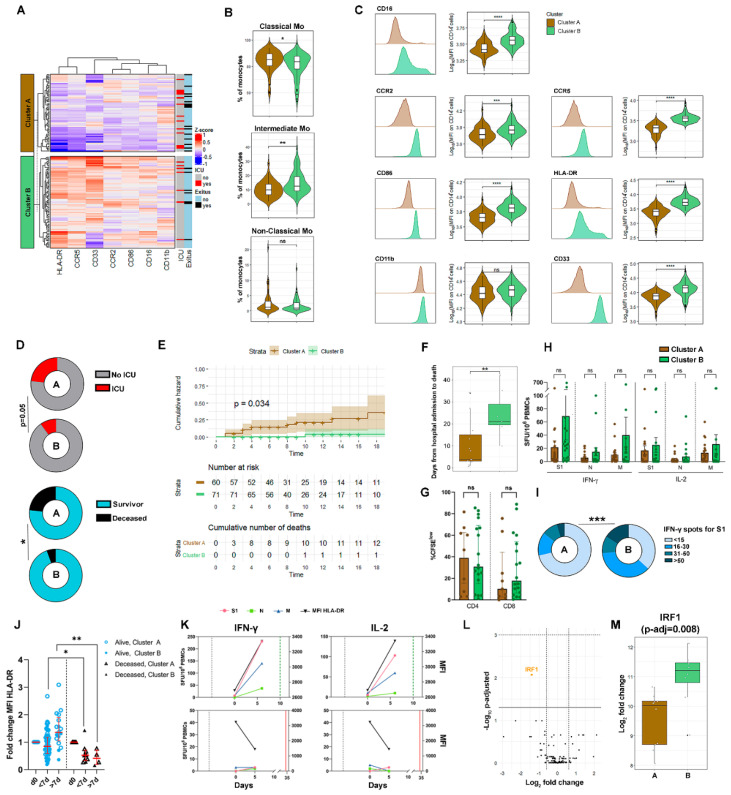
Acute COVID-19 patients with low expression of surface molecules in monocytes have higher risk of ICU requirement or death. (**A**) K-means clustering heatmap of monocyte surface markers from acute COVID-19 patients at ER arrival identified two separate clusters (**A**,**B**) of acute COVID-19 patients. (**B**) Cluster A acute COVID-19 patients showed a significantly higher proportion of classical monocytes, and lower proportion of intermediate monocytes, when compared to cluster B. (**C**) Comparison of monocyte surface markers between clusters. (**D**) Differences in ICU requirement and death rate in cluster A versus cluster B. (**E**) Cumulative death Kaplan-Meier analysis of hospitalized cluster A and B acute COVID-19 patients. (**F**) Interval from hospitalization to death in cluster A and B patients. (**G**) Polyclonal CD4 and CD8 T cell proliferation in cluster A and B patients. (**H**) IFN-γ and IL-2 specific T cell response against S1, N and M peptides of SARS-CoV-2 in cluster A and cluster B patients. (**I**) Cluster B patients had higher number of S1 specific T cells at ER arrival. (**J**) Follow-up of HLA-DR expression in monocytes during hospitalization showed augmentation in surviving patients and decrease in patients who died. (**K**) Follow-up of specific S1, N and M T cell response and HLA-DR expression on monocytes in two representative patients, a survivor (top) and a non-survivor (down). X-axes represent total days (Days of symptoms: <0; days of hospitalization: >0; Day 0 represents ER arrival day). Vertical black, dashed lines represent the day of symptom onset. Vertical red, solid line represents the day of dying. Vertical green, dashed line represents the day of discharge. (**L**) Volcano plot of differently up and downregulated genes between cluster A (*n* = 10) and cluster B (*n* = 8) monocytes. (**M**) IRF1 was significantly downregulated in cluster A. *, *p* < 0.05; **, *p* < 0.01; ***, *p* < 0.001; ****, *p* < 0.0001; ns, non-significant.

## Data Availability

Data have been deposited in the Gene Expression Omnibus (GEO). Transcriptomic data are available under accession number GSE180594. Epigenetic data are available under accession number GSE180523 (for the moment reviewers will have to enter the secure token cjgvqiuylfqtpyb, to view these data).

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
