# Peer review of "Alterations in Circulating Monocytes Predict COVID-19 Severity and Include Chromatin Modifications Still Detectable Six Months after Recovery"

_biomedicines, 2021, doi:10.3390/biomedicines9091253_

Round 1

Reviewer 1 Report

This is a very interesting study that attempts to identify long-lasting modification in monocytes after SARS-CoV-2 infection. The investigators showed that monocytes of COVID-19 patients showed phenotypical featurs (i.e. downregulation of HLA-DR) compared to monocytes isolated from HDs. Moreover, monocytes isolated from COVID-19 patients displayed the ability to release high amount of pro-inflammatory cytokines such as GM-CSF, TNFalfa, IL-6, IL-1b during infection, but at recovery they did not show this inflammation-associated imprinting. Moreover, the authors demonstrated that HLA-DR downregulation in monocytes was associated with higher levels of inflammation mediators and a lower T cell proliferation, suggesting that these virus-reprogrammed monocytes showed both immunosuppressive and pro-inflammation activity. The authors analyzed both the epigenetic status of monocytes during hospitalization and recovery time unveiling a conserved signature characterized by genes with inflammation-resolution and tissue-protection functions. To better understand this conserved gene profile, the authors assessed chromatin changes between acute and recovered infection by ATAC-seq analysis. The last part of the manuscript can not be evaluated since I did not find Figure 5 in the format.

Collectively, the results are of considerable interest but the authors should take in consideration some aspects:

  1. Figure 1C. The authors should analyze also some functional marker on monocytes such as the expression of PD-L1 as highlighted in Bost. et al. Deciphering the state of immune silence in fatal COVID-19 patients Nat. Communications, monocytes isolated from severe COVID-19 patients showed immunosuppressive features such as ARG1 and PD-L1 expression.
  2. Figure 1D. Data refer to LPS-stimulated monocytes. It would be helpful to also display data about unstimulated monocytes.
  3. Figure 2E. Data of proliferation of T cells need to be more characterized. What kind of cell subset was analyzed by the authors: naïve, effector, memory T cells?
  4. Figure 2F. The authors should test by multipletesting correction which is the most critical marker of monocytes associated with T cell proliferation block.
  5. The T cell proliferation block is mediated by cell-contact or soluble factors. In other words, are the monocyte-derived supernatants able to control in vitro T cell proliferation? This aspect if true needs to be discussed.
  6. Monocytes are short-lived cells in the blood. How the authors can conciliate the epigenetic imprinting in monocytes after 6 months? The authors probably should comment a “trained” immunity induced by the virus as possible explanation in the discussion.
  7. Data about Figure 5 can not be evaluated since I do not find this part in the attachment.

Minor point.

  1. Some relevant papers about monocytes/myeloid cells in COVID-19 patients need to be cited:

Bost. et al. Deciphering the state of immune silence in fatal COVID-19 patients Nat. Communications; demonstrates  the presence of immunosuppressive cells in COVID-19 patients characterized by ARG1 expression;

Reyes et al. Plasma from patients with bacterial sepsis or severe COVID-19 induces suppressive myeloid cell production from hematopoietic progenitors in vitro. Sci Transl Med; demonstrates that an in vitro short-term incubation of human hematopoietic stem and progenitor cells (HSPCs) with COVID-19 patient’s plasma promotes the generation of suppressive cells, with a strong expression of p-STAT3 and a dose-dependent up-regulation of IL6, IL10, TNF, IL1B genes;

Zhang et al. COVID-19 infection induces readily detectable morphological and inflammation-related phenotypic changes in peripheral blood monocytes, the severity of which correlate with patient outcome. J. Leukoc. Biol, demonstrates peculiar features of monocytes isolated from COVI-19 patients

  1. The quality of figures is modest

Author Response

Reviewer: 1

Major comments:

Question 1

Figure 1C. The authors should analyze also some functional marker on monocytes such as the expression of PD-L1 as highlighted in Bost. et al. Deciphering the state of immune silence in fatal COVID-19 patients Nat. Communications, monocytes isolated from severe COVID-19 patients showed immunosuppressive features such as ARG1 and PD-L1 expression.

Answer:

We agree with the reviewer that knowing the expression of immunosuppressive markers in monocytes of COVID-19 patients would be of great interest. Unfortunately those markers were not included in the flow cytometry panel depicted in Figure 1C. As we worked with fresh samples, studying these markers now would require recruiting a new patient cohort. This would be challenging as the number of patients who currently attend to the hospital with COVID-19 has dramatically dropped. Nevertheless, we will take this suggestion into consideration when planning future experiments. The Bost et al. paper was already cited in the original manuscript (old reference 28, current reference 30) but is again cited now in the Discussion section (please see Answer to Minor comment Question 1).

Question 2

Figure 1D. Data refer to LPS-stimulated monocytes. It would be helpful to also display data about unstimulated monocytes.

Answer:

All 11 cytokines were analyzed in supernatants from unstimulated monocyte cultures, however, in all cases cytokines were below the Luminex detection threshold and for that reason the results were not included in the figure.

In order to provide information on the cytokine production by unstimulated monocytes, we have added the following sentence in Figure 1 caption (page 7):   

In unstimulated monocyte wells all cytokines were below the detection threshold: GM-CSF<2.6 pg/mL; TNF-α<6.4 pg/mL; IL1-RA<1.6 pg/mL; IL1β<1.6 pg/mL; IL-6<0.64 pg/mL; IL-8<0.64 pg/mL; IL-10<2.6 pg/mL; IL-18<0.64 pg/mL; CCL2<3 pg/mL; CCL3<3 pg/mL; CXCL10<2.6 pg/mL.

Question 3

Figure 2E. Data of proliferation of T cells need to be more characterized. What kind of cell subset was analyzed by the authors: naïve, effector, memory T cells?

Answer:

Proliferation was recorded in CD4 and CD8 T cells, but no further T cell subsets were investigated.  We have specified the T cell subset in Figure 2E caption (page 9) as follows:

Proliferation of in vitro polyclonally stimulated CD4 and CD8 T cells from HC, acute COVID-19 and post-COVID-19.

Question 4

Figure 2F. The authors should test by multipletesting correction which is the most critical marker of monocytes associated with T cell proliferation block.

Answer:

As suggested by the reviewer, we have performed a multivariate correlation between monocyte surface markers expression and T cell proliferation. In the multivariate analysis we do not observe any statistically significant correlation and correlations between HLA-DR or CCR5 expression in monocytes and T cell proliferation are lost:

CD4 T cell proliferation

Coefficients

Estimate

Std. Error

P-value

CCR5

-7.305e-05

3.570e-02

0.8510

HLA-DR

1.341e-02

1.196e-02

0.3439

CD86

-4.879e-02

1.609e-02

0.0562

CCR2

-9.627e-04

6.921e-04

0.2584

CD11b

-1.627e-03

2.017e-03

0.4788

CD16

3.120e-02

1.585e-02

0.1437

CD33

2.209e-03

1.977e-03

0.3452

CD8 T cell proliferation

Coefficients

Estimate

Std. Error

P-value

CCR5

0.058833

0.055518

0.367

HLA-DR

-0.009945

0.018601

0.630

CD86

-0.025434

0.025017

0.384

CCR2

-0.001563

0.001076

0.242

CD11b

-0.006058

0.003136

0.149

CD16

0.015026

0.024647

0.585

CD33

-0.002503

0.003074

0.475

In our opinion this negative result could be due to the number of subjects tested in the proliferation assay being not large enough and in these conditions (multiple variables and low sample size), a multivariate analysis might not be adequate. Nevertheless, given these limitations, we have decided to move the panels in Figure 2F to the Supplementary Figure 3. See below the new version of Figure 2 and S3:

Figure 2. HLA-DR downregulation in monocytes correlates with higher plasma levels of IL-6, G-CSF, CXCL10, CCL2 and lower SARS-CoV-2-specific T cell response. (A) PCA of 20 cytokines in plasma from HC (n=16), acute (n=84) and post-COVID-19 (n=10) patients. Individual patient values (small points), mean (big point) and confidence interval of the mean (shadows) are represented. (B) Contribution of each cytokine to PCA dimensions 1 and 2. Dashed line represents the mean of all cytokine contributions. (C) Fold changes of median log-transformed values of plasma cytokines analyzed in HC, acute and post-COVID-19 patients. Dotted line represents unchanged cytokine level. (D) Correlation analysis between plasma cytokine levels and monocyte surface expression of CD11b, CCR2, CCR5, CD86, HLA-DR, CD16 and CD33. (E) Proliferation of in vitro polyclonally stimulated CD4 and CD8 T cells from HC, acute COVID-19 and post-COVID-19. (F) SARS-CoV-2 S1, N and M specific cellular response in HC (n=30), acute (n=40) and post-COVID-19 (n=41) patients. Dashed lines represent the established cut-off of positivity according to HC SFU means plus 3 standard deviations. (G) Correlation between specific T cell response and monocyte subsets and surface markers in acute and post-COVID-19 patients. Color of squares indicates direct (blue) or inverse (red) correlation. The size of the square is proportional to magnitude of correlation. *, p<0.05; **, p<0.01; ***, p<0.001; ****, p<0.0001.

Supplementary figure 3. Correlation between monocyte surface markers and CD4 and CD8 T cell proliferation in acute COVID-19 patients at ER arrival.

Question 5

The T cell proliferation block is mediated by cell-contact or soluble factors. In other words, are the monocyte-derive supernatants able to control in vitro T cell proliferation? This aspect if true needs to be discussed.

Answer:

In our work we have found an immunosuppressive phenotype in monocytes and a reduced T cell proliferation capacity during acute SARS-CoV-2 infection. Our data suggests that these two findings could be related but we have not experimentally tested if there is a direct association between them. Future studies to evaluate this association and elucidate its mechanism will be relevant.

In relation to this, we have added the following sentence in page 15 of the Discussion:

It will be relevant in future studies to elucidate if the changes detected in monocytes during acute COVID-19 are related with the observed limited T cell proliferation and through which mechanisms.

Question 6

Monocytes are short-lived cells in the blood. How the authors can conciliate the epigenetic imprinting in monocytes after 6 months? The authors probably should comment a “trained” immunity induced by the virus as possible explanation in the discussion.

Answer:

We agree with the reviewer that the changes observed in monocytes six months after infection could possibly reflect the “training” of monocytes through the reprogramming of myeloid progenitor cells triggered by SARS-CoV-2 infection. Following the suggestion of the reviewer, this hypothesis has been included in the Discussion (pages 15 and 16) as follows:

It has been shown that the long-lasting epigenetic changes observed in monocytes may correspond to a trained immunity mechanism caused by the capacity of different stimuli to reprogram myeloid progenitor cells [52]. Thus our results suggest that SARS-CoV-2 infection could induce trained immunity.

  1. Bekkering, S.; Dominguez-Andres, J.; Joosten, L.A.B.; Riksen, N.P.; Netea, M.G. Trained Immunity: Reprogramming Innate Immunity in Health and Disease. Annu Rev Immunol 2021, 39, 667-693, doi:10.1146/annurev-immunol-102119-073855.

Question 7

Data about Figure 5 cannot be evaluated since I do not find this part in the attachment.

Answer:

We apologize for the absence of Figure 5. It was included in the initial manuscript but we ignore why it was lost in the submission process. We have included it again in page 13 of the manuscript, and also here below in this document:

Figure 5. Acute COVID-19 patients with low expression of surface molecules in monocytes have higher risk of ICU requirement or death. (A) K-means clustering heatmap of monocyte surface markers from acute COVID-19 patients at ER arrival identified two separate clusters (A and B) of acute COVID-19 patients. (B) Cluster A acute COVID-19 patients showed a significantly higher proportion of classical monocytes, and lower proportion of intermediate monocytes, when compared to cluster B. (C) Comparison of monocyte surface markers between clusters. (D) Differences in ICU requirement and death rate in cluster A versus cluster B. (E) Cumulative death Kaplan-Meier analysis of hospitalized cluster A and B acute COVID-19 patients. (F) Interval from hospitalization to death in cluster A and B patients. (G) Polyclonal CD4 and CD8 T cell proliferation in cluster A and B patients. (H) IFN-γ and IL-2 specific T cell response against S1, N and M peptides of SARS-CoV-2 in cluster A and cluster B patients. (I) Cluster B patients had higher number of S1 specific T cells at ER arrival. (J) Follow-up of HLA-DR expression in monocytes during hospitalization showed augmentation in surviving patients and decrease in patients who died. (K) Follow-up of specific S1, N and M T cell response and HLA-DR expression on monocytes in two representative patients, a survivor (top) and a non-survivor (down). X-axes represent total days (Days of symptoms: <0; days of hospitalization: >0; Day 0 represents ER arrival day). Vertical black, dashed lines represent the day of symptom onset. Vertical red, solid line represents the day of dying. Vertical green, dashed line represents the day of discharge. (L) Volcano plot of differently up and downregulated genes between cluster A (n=10) and cluster B (n=8) monocytes. (M) IRF1 was significantly downregulated in cluster A. *, p<0.05; **, p<0.01; ***, p<0.001; ****, p<0.0001.

Minor comments:

Question 1

Some relevant papers about monocytes/myeloid cells in COVID-19 patients need to be cited: Bost et al Nat Communications; Reyes et al Sci Transl Med; and Zhang et al J. Leukoc Biol.

Answer:

We agree with the reviewer on the relevance of the above mentioned papers. The work by Bost et al. was already included in the submitted manuscript (old reference 28, current reference 30). We have now also added this reference in page 15 of the Discussion as follows:

Thus, an inflammation-resolution expression program characterizes the transcriptome of circulating monocytes during acute COVID-19 and may be related with the myeloid-driven immune suppression observed in other studies [30,51]. This program is still observed in convalescent phases of COVID-19.

According to the reviewer´s suggestion, the other two studies have been included in the manuscript.

Reyes et al, Sci Transl Med. 2021 appears in page 15 of the discussion with the new reference number 51:

Thus, an inflammation-resolution expression program characterizes the transcriptome of circulating monocytes during acute COVID-19 and may be related with the myeloid-driven immune suppression observed in other studies [30,51]. This program is still observed in convalescent phases of COVID-19.

Zhang D et al, J Leukoc Biol. 2021 appears in page 5 of the Results with the new reference number 23:

When compared to HC, monocytes from acute COVID-19 patients had higher size (FSC) and granularity (SSC), higher expression of CD11b, CD16 and CD33, similar expression of CCR2, CCR5 and CD86, and a marked downregulation of HLA-DR, as previously reported [11,14,23] (Figure 1C).

Question 2

The quality of the figures is modest

Answer:

We will work to improve the quality of the figures together with the Editorial Office.

Reviewer 2 Report

I read with the paper that aims to detect the alterations of monocytes in predicting gravity covid-19
however, there are some elements that the authors should clarify.
It is not clear whether the authors' goal is to verify whether the conditions found in the acute phase to predict the worst of the covid disease or whether the goal is to verify the damage at 6 months.
a particularly important data omitted by the authors is data to the disease phase at the time of enrollment (viral phase or inflammatory phase) (see 10.12998/wjcc.v8.i19.4280,  10.3390/biomedicines9080903; ) or to the days from the onset of symptoms.
this data is critical because it can be correlated to the correspondences data found.
they should also explain why the controls was different in number per cases (52 vs 45)
furthermore, it would be useful to correlate the basal alterations with other studies performed in the emergency room (see 10.1002/iid3.440)

a comparison with other studies on the subject would enrich the discussion (see 10.3389/fimmu.2021.720109)

Author Response

Reviewer: 2

Question 1

It is not clear whether the author´s goal is to verify whether the conditions found in the acute phase to predict the worst of the COVID disease or whether the goal is to verify the damage at 6 months.

Answer:

The objective of the present study is to understand the role of monocytes during SARS-CoV-2 infection, both during the acute phase of COVID-19 and during the recovery period. We realize that the absence of Figure 5 in the previous version of the manuscript may have undermined the clarity of the study’s goal. This figure has now been included.

Question 2

A particularly important data omitted by the authors is data to the disease phase at the time of enrollment (viral phase or inflammatory phase) (see 10.12998/wjcc.v8.i19.4280, 10.3390/biomedicines9080903) or to the days from the onset of symptoms. This data is critical because it can be correlated to the correspondences data found.

Answer:

Please see Supplementary Table 1 where we described patients’ clinical data at emergency room arrival (time of enrollment) including days from the onset of symptoms. The median days from illness onset to hospital admission was 7 days (IQR 4-10 days). There was no difference in the number of days from symptom onset between cluster A (more severe) and cluster B (less severe) patients. Based on the literature, these number of days would correspond with the viral phase of the disease. Despite the early arrival to the emergency room, it seems like some patients were already in the inflammatory phase of the disease. Evaluating monocyte characteristics at the emergency room or early upon admission could help to distinguish those patients with higher probabilities to develop a poor disease course.

Question 3

They should also explain why the controls was different in number per cases (52 vs 45)

Answer:

In this study we included three patient cohorts: healthy controls (n=45), acute COVID-19 patients (n=131) and post-COVID-19 patients (n=52).

The cohorts’ sizes are specified in page 2, Materials and Methods section:

We collected blood from 131 RT-PCR positive, acute phase COVID-19 patients (acute COVID-19), who attended the University Hospital 12 de Octubre ER with SARS-CoV-2 infection, from August 24 to November 5, 2020. In addition, we collected prospectively blood from acute COVID-19 patients at several times during their hospital stay. We also obtained samples from 52 patients who had overcome RT-PCR-confirmed COVID-19 six months before (Post-COVID-19). 45 healthy controls (HC) were also included.

And in page 5, Results section:

We enrolled 131 consecutive patients who attended the ER and were hospitalized with acute COVID-19. All patients had a positive real time polymerase chain reaction (RT-PCR) for SARS-CoV-2 (acute COVID-19 patients). In addition we obtained samples from another 52 patients 6 months after hospitalization because of positive RT-PCR, SARS-CoV-2 infection (post-COVID-19) and from 45 age- and gender-matched healthy controls (HC).

Some experiments were not performed in the entire patient cohort, mostly genetic experiments. This was specified in the text:

Page 9: We assessed the expression of 255 inflammation-related genes in freshly isolated monocytes from HC (N=7), early acute (N=18) and post-COVID-19 patients (N=8).

Page 11: ... the genome-wide accessibility of monocyte chromatin was analyzed in three patients from each cohort by ATAC-Seq.

Question 4

Furthermore, it would be useful to correlate the basal alterations with other studies performed in the emergency room (see 10.1002/iid.440)

Answer:

The reference mentioned by the reviewer has been included (reference number 8):

  1. Ucciferri, C.; Caiazzo, L.; Di Nicola, M.; Borrelli, P.; Pontolillo, M.; Auricchio, A.; Vecchiet, J.; Falasca, K. Parameters associated with diagnosis of COVID-19 in emergency department. Immun Inflamm Dis 2021, 9, 851-861, doi:10.1002/iid3.440.

and cited in the Introduction section (page 1) as follows:

Higher C-reactive protein, neutrophil-to-lymphocyte ratio, lactate dehydrogenase and interleukin (IL)-6 are normally found in severe COVID-19 and predict fatal outcome [5-8]  

Question 5

A comparison with other studies on the subject would enrich discussion (see 10.3389/fimmu.2021.720109).

Answer:

The reference mentioned by the reviewer has been included (reference number 32):

  1. Knoll, R.; Schultze, J.L.; Schulte-Schrepping, J. Monocytes and Macrophages in COVID-19. Front Immunol 2021, 12, 720109, doi:10.3389/fimmu.2021.720109.

and cited in the Discussion section (page 14) as follows:

We confirmed that in comparison to HC, in early acute COVID-19 non-classical monocytes were reduced while classical and intermediate monocytes predominated and showed downregulation of HLA-DR surface expression [1,2,4-8].

Round 2

Reviewer 1 Report

The authors ansewred all my concerns. This article is highly relevant to the field immunology of COVID-19, it is well-done and well-written. 

Author Response

Thank you very much for your comments.

Reviewer 2 Report

the authors responthe authors responded sufficiently to the objectionsded sufficiently to the objections
I suggest adding the citations (10.12998/wjcc.v8.i19.4280, 10.3390/biomedicines9080903 )or similar in the text as already indicated

Author Response

Reviewer: 2

The authors responded sufficiently to the objections. I suggest adding the citations (10.12998/wjcc.v8.i19.4280, 10.3390/biomedicines9080903) or similar in the text as already indicated.

Answer:

According to the reviewer´s suggestion, the two studies have now been included in the manuscript.

Ucciferri et al, Worl J Clin Cases 2020 (10.12998/wjcc.v8.i19.4280) appears in page 1 of the discussion with the new reference number 11:

Important disturbances in the antiviral immune response characterize the severe forms of acute COVID-19, as SARS-CoV-2 evades the innate immunity by impairing the interferon (IFN) type I and III responses, and causes hyperinflammation with increased production of cytokines and chemokines [9-11].

Nappi et al, biomedicines 2021 (10.3390/biomedicines9080903) appears in page 13 of the discussion with the new reference number 31:

Remarkably, known factors associated with a critical course of the infection, such as age, sex, comorbidities, ARDS, ratio of peripheral blood oxygen saturation to fraction of inspired oxygen (SpO2/FiO2), monocytes, platelets, neutrophil-to-lymphocyte ratio, LDH, albumin, AST, ALT, fibrinogen, D-dimer, viral load (Ct) [28-31], days of symptoms and treatment received were not different between clusters.